# Estimated Costs Associated with Surgical Site Infections in Patients Undergoing Cholecystectomy

**DOI:** 10.3390/ijerph19020764

**Published:** 2022-01-11

**Authors:** Natividad Algado-Sellés, Javier Mira-Bernabeu, Paula Gras-Valentí, Pablo Chico-Sánchez, Natali Juliet Jiménez-Sepúlveda, Marina Fuster-Pérez, José Sánchez-Payá, Elena María Ronda-Pérez

**Affiliations:** 1Epidemiology Unit, Preventive Medicine Service, Alicante General University Hospital, 03010 Alicante, Spain; algado_nat@gva.es (N.A.-S.); gras_pau@gva.es (P.G.-V.); chico_pab@gva.es (P.C.-S.); jimenez_natsep@gva.es (N.J.J.-S.); fuster_marper@gva.es (M.F.-P.); 2Alicante Institute for Health and Biomedical Research (ISABIAL), 03010 Alicante, Spain; 3Preventive Medicine Service, General University Hospital of San Juan de Alicante, 03550 Sant Joan d’Alacant, Spain; mira_javiber@gva.es; 4Area of Preventive Medicine and Public Health, Faculty of Health Sciences, University of Alicante, 03690 Alicante, Spain; elena.ronda@ua.es; 5Centre of Networked Biomedical Research in Epidemiology and Public Health (CIBERESP), 28029 Madrid, Spain

**Keywords:** cholecystectomy, surgical wound infection, healthcare-associated infections, cost analysis, healthcare costs

## Abstract

Among healthcare-associated infections, surgical site infections (SSIs) are the most frequent in Spain. The aim of this work was to estimate the costs of SSIs in patients who underwent a cholecystectomy at the Hospital General Universitario de Alicante (Spain) between 2012–2017. This was a prospective observational cohort study. The Active Epidemiological Surveillance Program at our hospital recorded all the cholecystectomies performed. Risk factors associated with the development of SSIs were determined by multivariate analysis and two homogeneous comparison groups were obtained by using the propensity score. The number of extra days of hospital stay were recorded for patients with an SSI and with the cost per hospitalised day data, the additional cost attributed to SSIs was calculated. A total of 2200 cholecystectomies were considered; 110 patients (5.0%) developed an SSI. The average length of hospital stay was 5.6 days longer among patients with an SSI. The cost per SSI was EUR 1890.60 per patient, with the total cost for this period being EUR 207,961.60. SSIs after cholecystectomy lead to a prolongation of hospital stay and an increase in economic costs. It is essential to implement infection surveillance and control programs to reduce SSIs, improve patient safety, and reduce economic burden.

## 1. Introduction

Healthcare-associated infections (HAIs) are defined as infections contracted during a patient’s stay in hospital or at another health centre and which the patient did not have at the time of admission [1]. HAIs are currently the most common complication among hospitalised patients [2] and are a recognised cause of morbidity and mortality [3], prolonged hospital stays, and huge avoidable additional costs for healthcare systems [4,5,6]. Thus, they represent a major health problem and a challenge to public health systems around the world. 

A 2018 study by the European Centre for Disease Prevention and Control (ECDC) on the prevalence of HAIs (2016–2017) [7], revealed that 5.9% of hospitalised patients developed a HAI, with the prevalence ranging from 4.4% to 7.1%. According to this study, the most frequent HAIs were respiratory (25.7%), followed by urinary (18.9%), surgical site (18.4%), and bacteraemia (10.8%) [7]. At the national level, the 2018 Study of Prevalence of Nosocomial Infections in Spain (EPINE-EPPS) [8] estimated that 7.2% of hospitalised patients presented some form of HAI, with surgical site infections (SSIs) being the most frequent (27.2%). In our centre, the General University Hospital of Alicante (HGUA), the prevalence of HAIs in 2018 was 7.16%.

The risk of developing an SSI depends on several factors including: antibiotic prophylaxis, skin preparation with an antiseptic solution, the NNISS index, the duration of the operation, the degree of contamination of the surgery, the urgency of the operation, the surgical approach (open or laparoscopic surgery) and the surgical team. Other factors that may be involved are normoglycemia and normothermia. Importantly, patients who develop an SSI are 60% more likely to be admitted to critical care units, twice as likely to die, and five times more likely to be readmitted [9]. In turn, SSIs entail a prolongation of the hospital stay and an increased number of tests, procedures, and antimicrobial usage [10]. Thus, SSIs carry a significant economic burden, both for patients and the healthcare system, with the need for extended hospitalisations being the most important factor contributing to increased costs [11]. Despite this, as not all SSIs have the same characteristics, the failure to perform analyses stratified according to the surgical procedure type can lead to erroneous conclusions about prolonged hospital stays attributed to SSIs [11,12,13].

For example, infection of the surgical wound after cardiac surgery increases the hospital stay by an average of 20 days, amounting to an additional EUR 20,000 in hospital costs per case, while SSIs following orthopaedic surgeries can extend the stay by a mean of 14 days [14]. Thus, variability related to the development of SSIs depends on the underlying surgery type, and for this reason, it is essential that the analysis of the costs associated with SSIs be carried out on a case-by-case basis for each type of surgical procedure in order to obtain more precise data on the impact of these infections according to the type of procedure being analysed. Given the fact that each procedure has its own characteristics, we should be aware that the results obtained from the analysis of one procedure cannot be applied to other surgical interventions. Nonetheless, very few studies have individually examined the costs associated with SSIs for each surgery type. Among other reasons, the difficulty in obtaining case series sufficiently large for analysis for some types of surgeries is likely.

Among the procedures encompassed by general and digestive system surgery, cholecystectomy (CCE) is one of the most frequent abdominal surgeries in the world. Since 1990, laparoscopy has been the approach of choice for this surgery as it carries a low risk of complications [14], with the risk dramatically increasing when open surgery is performed [15]. A meta-analysis on the use of antibiotic prophylaxis in laparoscopic CCEs observed that the incidence of SSIs after this type of surgery was 2.4% [16]. To the best of our knowledge, no previous studies have estimated the cost of infections in CCE surgical wounds. Based on the hypothesis that patients who developed an SSI after cholecystectomy had a longer hospital stay than those who did not become infected, the aim of this study was to estimate the costs associated with the development of SSIs after cholecystectomy using days of prolonged hospital stay as the unit of measurement.

## 2. Materials and Methods

We carried out a prospective cohort-type observational study at the General University Hospital of Alicante (HGUA). All patients who had undergone CCE surgery between 1 January 2012 and 31 December 2017 who had a postsurgical stay of more than 24 h were included and both urgent and scheduled surgeries were considered.

We used the criteria established by the Centers for Disease Control and Prevention (CDC) to define and classify SSIs [17]. The outcome variable was defined as the hospital stay length and the presence of an SSI as an explanatory variable.

In addition, the following variables were also recorded: sex, age, preoperative stay time, preoperative blood glucose, cut/shaved hair, laparoscopic technique, ASA (American Society Anesthesiologists) anaesthetic risk classification (where I = healthy, II = mild systemic disease, III = severe systemic disease, IV = disabling systemic disease, and V = terminal status), intervention time (minutes), type of surgery (dirty, clean/contaminated), and NNISS index (National Nosocomial Infections Surveillance Systems) to predict the risk of SSI occurrences, urgent interventions, antibiotic prophylaxis, and year.

For antibiotic prophylaxis assessment, the protocols approved by the Centre’s Infections Commission were used as the reference standard. The NNISS index comprises three risk factors: the highest ASA classifications (III, IV, and V), degree of surgical contamination (dirty or contaminated), and an intervention duration (in minutes) greater than the 75th percentile of the duration expected for that procedure. The presence of each of these factors adds one point to the NNISS score while one point is subtracted for laparoscopic surgeries, with the final score for each intervention ranging from 0 to 3 points [10].

Patient information was obtained from their digital medical histories (Orion Clinic), microbiology program (GestLab), and pharmaceutical prescription program for hospitalised patients (Prisma). Since 1991, the Preventive Medicine Service at the HGUA has been developing its Program for the Surveillance, Prevention, and Control of Healthcare-Associated Infections, which includes the surveillance of SSIs for 12 surgical procedures. Data collection was carried out by the nursing staff at this service and was supervised by its medical staff. A specific form designed for the epidemiological surveillance of SSIs was used to record these data. Patients were followed up for 30 days to determine whether they had developed an SSI. For patients who developed an SSI, the type of infection (superficial, deep, or in the organ space), whether a surgical wound culture had been requested, the type of microorganism isolated in the culture and the antibiotic used to treat the infection were also collected. Cholecystectomised patients who developed an SSI were defined as ’exposed’ and those who did not develop an SSI were termed ’unexposed’.

### Statistical Analysis

A bivariate logistic regression analysis was carried out to check whether the risk factors for SSIs described in the literature were associated with the development of this type of infection. Chi-squared tests were used to determine the association of these factors with the development of an SSI, while odds ratios (ORs) were obtained using a logistic regression model with 95% confidence intervals (95% CIs) to calculate the magnitude of the association; the statistical significance level was set to 5% (a *p*-value less than 0.05). To assess the independence of SSI risk factors from each other, variables with statistically significant associations were included in a multivariate logistic regression analysis that determined the relationship between them to identify the set of variables that best explained the probability of developing an SSI.

The adjusted OR was calculated using a logistic regression model with 95% CI and the individual probability of developing an SSI (propensity score, or PS) was obtained from these data [18,19]. The PS is a matching technique used to obtain two comparison groups which are differentiated only by the result variable (in this case, SSIs). Thus, its use of controls for potential confounding variables in observational studies by obtaining homogeneous groups with which levels of evidence similar to those of experimental studies can be achieved [11,20,21]. All the statistical analyses were performed using SPSS software (version 25.0; IBM Corp., Armonk, NY, USA).

Once the two comparison groups were obtained, the excess hospital stay time associated with SSIs (outcome variable) was calculated as the difference in days in postoperative stay between infected and matched non-infected patients (SSI: explicative variable). Student *t*-tests were used to calculate whether the differences in the mean length of stay between the exposed and unexposed patients was significant. According to data provided by the Economic Directorate at our centre, the cost per day of general surgery hospital stay was EUR 377.60. Therefore, we multiplied this figure by the number of extra days of hospital stay attributed to SSIs.

## 3. Results

We included 2200 patients who had undergone a cholecystectomy, of which, 110 (5.0%) developed SSIs. The incidence of SSIs was higher when an open surgery technique had been used, with 47 patients from a total of 248 open surgery interventions (19.0%) presenting an SSI compared with 63 from 1952 (3.2%) after undergoing a laparoscopic technique. Table 1 shows the risk factors associated with SSIs; age, preoperative blood glucose, laparoscopic technique, intervention duration, degree of surgical contamination, and the NNISS index score were all significantly related to the occurrence of SSIs.

The patient characteristics, before adjustment and after matching by PS, are shown in Table 2. The PS included in the multivariate logistic regression analysis showed those variables that were statistically significantly associated with the development of surgical site infection. These variables were: age, pre-surgical glycemia, laparoscopic technique, time of the intervention, type of surgery and NNISS index. Of note, before PS was applied there was no homogeneity between the two groups, with significant differences found between them. However, once PS had been applied, the groups were homogeneous, and no significant differences were found between them except in terms of the frequency of the laparoscopic technique.

The impact of SSIs on hospital stay length and associated costs is described in Table 3. The unexposed comparison group was obtained after applying the Propensity Score. The mean hospital stay length was 8.4 ± 11.1 days in the group of patients who had developed SSIs, which was a mean 5.6 days longer than those without SSIs. An intervention duration exceeding 80 min and preoperative blood glucose levels in excess of 200 mg/dL were associated with an average of 6.7 days extra hospital stay length in both cases, while the NNISS score was associated with a mean increase in hospital stay length of 6.6 days. The mean cost of each SSI case was EUR 1890.60, with a total cost of EUR 207,961.60 for all patients included in this study.

The most commonly used antibiotics in the treatment of SSIs after cholecystectomy were amoxicillin/clavulanic acid (34.55%; 38/110 patients), tazocel (28.18%; 31/110 patients) and meropenem (20.91%; 23/110 patients). In 14.55% (16/110) of patients with an SSI, there were no data on the antibiotic treatment used.

As for the microorganisms responsible for SSIs, of the 110 patients who developed infection, 61.8% (68/110) had a surgical wound culture requested, which was positive in 88.2% (60/68) of cases. The pathogens most frequently identified in the microbiological cultures were Escherichia coli (35%; 21/60 cultures) and Enterococcus (30%; 18/60). Regarding the genus Enterococcus, the species identified were E. faecalis (8/18) and E. faecium (5/18), in the remaining 5 cases the laboratory results did not indicate the species.

## 4. Discussion

The impact of SSIs on prolonging hospital stays and their associated costs have been demonstrated by numerous studies [22,23]. Most previous works have examined the global costs secondary to SSIs and simultaneously included multiple surgical procedures in their analyses. For example, Lissovoy [11] estimated that the cost of SSIs was USD 20,842 per patient for all surgical categories. However, the analysis of costs secondary to SSIs must be stratified by procedure type as both infection rates and durations of hospital stay vary depending on the surgical procedure considered [24,25,26].

Among the procedures included within the general surgery category, CCE is one of the most frequent surgical interventions in all hospitals and, in turn, is associated with a low incidence of SSIs [16,27]. Perhaps a reason we were unable to find any studies that specifically estimated the costs of SSIs after CCEs was the difficulty in reaching a sample size large enough to allow this analysis to be carried out [28].

This current study analysed the costs exclusively related to SSIs after CCEs and used the PS as a matching method to obtain homogeneous comparison groups. Our analysis revealed that over a 7-year follow-up period, SSIs after CCEs prolonged patient hospital stays by an average of 5.6 days at an additional cost of a mean EUR 207,961.60. The increase in length of stay is of a particularly significant value as it leads to a delay in scheduled admissions as well as an increase in the cost of care that could be invested in other items. However, we could not compare this data with similar studies on SSIs for this procedure as we were unable find any published work in this regard in the academic literature.

One of the limitations of this study is that normoglycemia, non-normothermia and surgical teams were not included among the risk factors for the development of SSI. We consider that this would be an aspect to be considered in future studies analysing the costs associated with SSIs. Another limitation of the study was the fact that after the application of PS with the aim of obtaining two homogeneous comparison groups, it was not possible to make this adjustment with respect to the laparoscopic technique variable, thus there were significant differences with respect to the surgical technique between the exposed and non-exposed groups.

We used excess days of hospitalisation as a surrogate measure of economic cost in this work. However, this study may have underestimated the true economic cost of SSIs in CCEs because the cost per hospitalised day only included personnel costs plus general expenses but omitted direct costs such as reinterventions, diagnostic tests, and therapeutic or pharmacy expenses. Moreover, another reason this cost may have been underestimated was because we did not consider whether any of the days during the hospitalisation of these patients included an admission to a critical care unit [23]; these units incur significantly higher costs per day relative to the cost of hospitalisation in surgical service units. Another aspect that may have underestimated the cost of SSIs is the failure to collect the doses and duration of antibiotic therapy used to treat infections.

HAI surveillance and control programs are great tools to minimise the appearance and economic impact of SSIs. Adherence to these programs constitute a crucial element within the framework of clinical safety and should be prioritised in policies designed to maximise both the quality of the healthcare services rendered and patient safety [29]. Thus, their implementation and compliance with these measures should be prioritised in every healthcare system to reduce both morbidity and mortality rates and economic costs associated with SSIs.

By carrying out stratified analyses in future studies of SSIs, researchers will be able to better evaluate the effectiveness of infection prevention and control programs and propose intervention strategies to reduce SSIs within this type of surgical procedure to plan new strategies that offer high-quality healthcare while also maximising patient safety.

Of note, this study made it possible to estimate the cost associated with SSIs in patients who had undergone CCEs in our healthcare setting and based on current data. This latter factor must be considered since both the surgical techniques used and their associated costs differ according to each individual healthcare system and often change over time, meaning that studies analysing current data are crucial. Moreover, in future studies this current estimate will contribute to calculating the cost–benefit ratio of SSI surveillance and prevention programs for surgical procedures at the HGUA and thus, will help evaluate the benefits of the implementation and compliance with such programs.

## 5. Conclusions

This study allows us to appreciate the costs attributed to SSI after cholecystectomy, one of the most frequent surgical techniques within general surgery. Starting from homogeneous comparison groups and measured in terms of length of hospital stay, this study provides us with data on the individual cost of each SSI after cholecystectomy, data that have not been found published in the reviewed literature. These results help us, on the one hand, to evaluate the efficiency of existing SSI surveillance, prevention, and control programs at the present time and, derived from this, to be able to plan new policies to optimise the resources allocated to health care. This study can be compared with others carried out in countries with a public health system such as ours, where the costs are comparable; however, these costs may be different for other health system models. What can be extrapolated from studies carried out in countries with other health care models is the methodology used, which guarantees that the variables included present a homogeneous distribution in both comparison groups and is a novel aspect in the field of cost studies.

## Figures and Tables

**Table 1 ijerph-19-00764-t001:** Factors associated with surgical site infection after cholecystectomy.

	SSI ^a^ % (*n*) *N* = 2200	OR ^b^ (95% CI)	P ^c^	ORa ^d^ (95% CI)	Pa ^e^	ORa ^f^ (95% CI)	Pa ^e^
**Sex**							
Male	6.3 (55/878)	1.5 (1.0–2.3)	0.027	1.0 (0.7–1.6)	0.934	1.1 (0.7–1.7)	0.604
Female	4.2 (55/1322)	1		1		1	
**Age**							
>65 years	7.6 (69/905)	2.5 (1.7–3.7)	<0.001	1.8 (1.1–2.8)	0.015	1.7 (1.1–2.7)	0.010
≤65 years	3.2 (41/1295)	1		1		1	
**Presurgical stay**							
≥1 day	5.4 (46/858)	1.3 (0.8–1.7)	0.530	-	-	-	-
0 days	4.8 (64/1342)	1					
**Pre-surgical blood glucose levels ***							
>200 mg/dL	14.6 (6/41)	3.6 (1.9–8.8)	0.030	2.1 (0.8–5.3)	0.129	2.4 (1.0–5.9)	0.048
<200 ms/dL	4.5 (93/2049)	1		1		1	
**Cut/shaved hair**							
Yes	4.9 (37/752)	1.0 (0.7–1.5)	0.970	-	-	-	-
No	4.9 (70/1434)	1					
**Laparoscopic technique**							
No	19.0 (47/248)	7.0 (4.7–10.5)	<0.001	4.4 (2.8–6.9)	<0.001	-	-
Yes	3.2 (63/1952)	1		1			
**ASA**							
ASA 3–4	9.0 (35/388)	2.3 (1.5–3.5)	<0.001	1.4 (0.7–2.7)	0.368	-	-
ASA 1–2	4.1 (75/1812)	1		1			
**Intervention time (min)**							
>80 min	10.7 (52/484)	3.4 (2.3–5.1)	<0.001	2.1 (1.3–3.2)	0.002	-	-
≤80 min	3.4 (58/1716)	1		1			
**Surgery type**							
Dirty	8.6 (53/617)	2.5 (1.7–3.7)	<0.001	2.0 (1.0–3.9)	0.039	-	
Clean/contaminated	3.6 (57/1583)	1		1			
**NNISS Index score**							
NNISS 0	2.5 (28/1115)	1		-	-	-	-
NNISS 1	5.2 (34/651)	2.1 (1.3–3.6)	0.003			2.2 (1.2–3.8)	0.003
NNISS 2	9.4 (33/352)	4.0 (2.4–6.7)	<0.001			4.5 (2.4–8.4)	<0.001
NNISS 3	18.3 (15/82)	8.7 (4.4–17.1)	<0.001			9.5 (4.1–22.2)	<0.001
**Urgent intervention**							
Yes	6.6 (36/542)	1.5 (1.0–2.3)	0.043	1.5 (0.7–2.9)	0.270	1.9 (1.0–3.5)	0.038
No	4.5 (74/1658)	1		1		1	
**Antibiotic prophylaxis**							
Indicated: implemented and adequate.	3.7 (52/1416)	1		1		1	
Indicated: implemented and inadequate.	8.3 (13/156)	2.4 (1.3–4.5)	0. 007	1.6 (0.8–3.2)	0.209	1.9 (1.0–3.9)	0.046
Indicated: not implemented.	5.2 (13/252)	1.4 (0.8–2.7)	0.264	1.7 (0.9–3.3)	0.099	1.6 (0.8–3.0)	0.165
Not indicated.	8.5 (32/376)	2.5 (1.6–3.9)	<0.001	0.9 (0.4–1.8)	0.738	1.2 (0.6–2.3)	0.661
**Year**							
2012	6.0 (23/385)	1		-	-	-	-
2013	4.2 (15/355)	0.7 (0.4–1.4)	0.284				
2014	4.3 (17/396)	0.7 (0.4–1.3)	0.289				
2015	5.6 (22/395)	0.9 (0.5–1.7)	0.809				
2016	5.3 (17/322)	0.9 (0.5–1.7)	0.691				
2017	4.6 (16/347)	0.8 (0.4–1.5)	0.414				

^a^ SSI: surgical site infection; ^b^ OR: crude odds ratio; ^c^ P: *p*-value; ^d^ ORa: odds ratio adjusted for sex, age, preoperative blood glucose, laparoscopic technique, ASA, intervention time, surgery type, urgent intervention, antibiotic prophylaxis, and year; ^e^ Pa: adjusted *p*-value; ^f^ ORa: odds ratio adjusted for sex, age, preoperative blood glucose, NNISS index, urgent intervention, antibiotic prophylaxis, and year; * 99 cases were lost with respect to the preoperative blood glucose variable and so *n* = 2101.

**Table 2 ijerph-19-00764-t002:** Patient characteristics before and after matching using the propensity score.

	BEFORE			AFTER		
	SSI ^a^ (*n* = 110) % (*n*)	No SSI (*n* = 2090) % (*n*)	P ^b^	SSI ^a^ (*n* = 110) % (*n*)	No SSI (*n* = 110) % (*n*)	P ^b^
**Sex** (Male)	50.0 (55)	39.4 (823)	0.027	50 (55)	51.8 (57)	0.787
**Age** (>65 years)	62.7 (69)	40.0 (836)	<0.001	62.7 (69)	55.5 (61)	0.273
**Presurgical stay** (≥1 day)	41.8 (46)	38.9 (812)	0.534	41.8 (46)	34.5 (38)	0.267
**Glycemia Pred ^c^** (>200 mg/dL)	6.4 (7)	1.8 (35)	0.001	6.4 (7)	4.5 (5)	0.553
**Cut/shaved hair** (Yes)	36.4 (40)	34.7 (726)	0.727	36.4 (40)	42.7 (47)	0.334
**Laparoscopic technique** (No)	42.7 (47)	9.6 (201)	<0.001	42.7 (47)	12.7 (14)	<0.001
**ASA** (ASA 3–4)	89.1 (98)	76.7 (1604)	0.003	89.1 (98)	84.5 (93)	0.319
**Surgery duration** (>80 min)	47.3 (52)	20.7 (432)	<0.001	47.3 (52)	37.3 (41)	0.133
**Type of surgery** (Dirty)	48.2 (53)	27.0 (564)	<0.001	48.2 (53)	43.6 (48)	0.499
**NNISS**			<0.001			0.501
NNISS 0	25.5 (28)	52.0 (1087)		25.5 (28)	33.6 (37)	
NNISS 1	30.9 (34)	29.5 (617)		30.9 (34)	25.5 (28)	
NNISS 2	30.0 (33)	15.3 (319)		30.0 (33)	25.5 (28)	
NNISS 3	13.6 (15)	3.2 (67)		13.6 (15)	15.5 (17)	
**Urgent intervention** (Yes)	32.7 (36)	24.2 (506)	0.043	32.7 (36)	33.6 (37)	0.886
**Antibiotic prophylaxis**			<0.001			0.151
Indicated: implemented and adequate.	47.3 (52)	65.3 (1364)		47.3 (52)	49.1 (54)	
Indicated: implemented and inadequate.	11.8 (13)	6.8 (143)		11.8 (13)	3.6 (4)	
Indicated: not implemented.	11.8 (13)	11.4 (239)		11.8 (13)	13.6 (15)	
Not indicated.	29.1 (32)	16.5 (344)		29.1 (32)	33.6 (37)	
**Year**			0.840			0.546
2012	20.9 (23)	17.3 (362)		20.9 (23)	14.5 (16)	
2013	13.6 (15)	16.3 (340)		13.6 (15)	17.3 (19)	
2014	15.5 (17)	18.1 (375)		15.5 (17)	16.4 (18)	
2015	20.0 (22)	17.8 (373)		20.0 (22)	15.5 (17)	
2016	15.5 (17)	14.6 (305)		15.5 (17)	14.5 (16)	
2017	14.5 (16)	15.8 (331)		14.5 (16)	21.8 (24)	

^a^ SSI: surgical site infection; ^b^ P: *p*-value; ^c^ pre-operative blood glucose levels.

**Table 3 ijerph-19-00764-t003:** Analysis of the costs associated with the increased postoperative stay related to surgical site infections.

	**Stay SS ^a^** **’Exposed’** **(µ ^b^ ± σ ^c^)**	**Stay SS ^a^** **‘Not Exposed’** **(µ ^b^ ± σ ^c^)**	**Stay Difference (Days)**	**Extra Cost per Patient (× €337.6)**	**Total Extra Cost in Euros (*n*)**
**Total**	8.4 ± 11.1	2.8 ± 2.7	5.6	1890.60	207,961.60 (110)
**Sex**					
Male	9.8 ± 12.2	3.4 ± 3.1	6.4	2160.60	118,835.20 (55)
Female	6.9 ± 9.7	2.1 ± 2.1	4.8	1620.50	89,126.40 (55)
**Age**					
>65	9.4 ± 12.0	3.4 ± 3.1	6.0	2025.60	1,397,664.40 (69)
≤65	6.5 ± 9.3	2.0 ± 1.9	4.5	1519.20	62,287.20 (41)
**Pre-surgical blood glucose levels ***					
>200 mg/dL	12 ± 9.7	5.4 ± 4.7	6.7	2261.90	13,571.50 (6)
<200 mg/dL	8.1 ± 11.2	2.6 ± 2.6	5.5	1856.80	172,682.40 (93)
**LT ^d^**					
No	11.4 ± 13.0	6.2 ± 3.7	5.2	1755.50	82,509.40 (47)
Yes	6.1 ± 8.9	2.8 ± 2.2	3.3	1114.10	70,187.00 (63)
**ASA ^e^**					
>1	8.5 ± 11.4	2.4 ± 3.6	6.1	2059.40	72,077.60 (35)
0–1	6.8 ± 9.0	1.6 ± 0.9	5.2	1755.50	131,664.00 (75)
**Intervention time**					
>80 min	10.2 ± 10.5	3.5 ± 4.1	6.7	2261.90	117,618.80 (52)
≤80 min	6.7 ± 11.5	2.0 ± 2.1	4.7	1586.70	92,029.80 (58)
**Surgery type**					
Dirty	11.0 ± 10.6	4.6 ± 3.0	5.4	1823.00	96,621.10 (53)
Clean/contaminated	5.9 ± 11.1	1.4 ± 1.5	4.5	1519.20	86,594.40 (57)
**NNISS Index score**					
NNISS 2–3	11.1 ± 10.9	4.5 ± 3.1	6.6	2228.20	106,951.70 (48)
NNISS 0–1	6.2 ± 10.9	1.6 ± 1.6	4.6	1552.90	96,283.50 (62)
**Year**					
2012	9.6 ± 11.1	3.9 ± 3.6	5.7	1924.30	44,259.40 (23)
2013	11.8 ± 13.6	2.3 ± 1.6	9.5	3207.20	48,108.00 (15)
2014	11.9 ± 15.4	4.6 ± 4.0	7.3	2464.50	41,896.20 (17)
2015	5.1 ± 6.7	2.0 ± 1.8	3.1	1046.60	23,024.30 (22)
2016	7.4 ± 11.6	1.0 ± 1.8	5.5	1856.80	31,565.60 (17)
2017	5.1 ± 5.4	2.1 ± 1.9	3.0	1012.80	16,204.80 (16)

^a^ SS: surgical services; ^b^ µ: mean; ^c^ σ: standard deviation; ^d^ LT: laparoscopic technique; ^e^ ASA: anaesthetic risk score. * There were 99 lost cases and so, *n* = 2101.

## Data Availability

The data presented in this study are available on request from the corresponding author. The data are not publicly available.

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
