# Peer review of "Estimated Costs Associated with Surgical Site Infections in Patients Undergoing Cholecystectomy"

_ijerph, 2022, doi:10.3390/ijerph19020764_

Round 1
Reviewer 1 Report
This is a second review I am providing, having reviewed a prior version of the manuscript. It appears that the authors have addressed the major issues that were the concern of this reviewer, particularly the authors should clearly acknowledge and state the fact that surgical site infection rates are different depending on the preoperative factors, the nature of the intraoperative processes, and the quality of postoperative care. Besides, there are factors related to care providers/surgical team, patient factors, and factors related to the type of operative procedure and the availability of advanced monitoring equipment. These study findings could be useful at least for local consumption in the healthcare facilities where the study was based although the external validity of such a study is uncertain. Further review is left to the discretion of the authors.
Author Response
Response:
Regarding surgery-related factors the study included ASA anesthetic risk, NNISS index, duration of surgery, degree of contamination of surgery, urgency of surgery as well as route of approach (open or laparoscopic surgery). The adequacy of antibiotic prophylaxis was also evaluated. Regarding skin preparation with antiseptic solution, this variable was not included in the analysis because it was carried out in 100% of the cases included in the study.
Normoglycemia and normothermia were not evaluated, constituting a limitation of this study in relation to the risk factors for surgical wound infection (SSI) evaluated. The surgical team was not evaluated either, this being another limitation. We consider that these three variables should be included in future studies analyzing the costs associated with SSI in order to assess their role in the occurrence of SSI.
Regarding the validity of the study, it could be compared with other studies carried out in countries with a public health system such as ours, where the costs are comparable; however, these costs may differ when faced with other health system models. In studies carried out in countries with other health care models, the methodology used in this study can be extrapolated, which has used a matching method that guarantees that the variables included present a homogeneous distribution in the group of patients with infection and the patients who have not developed infection with whom they have been matched. From the two comparison groups obtained, the difference in length of stay between the two is calculated.

Reviewer 2 Report
no further comments
Author Response
We thank the reviewer for his work.

Reviewer 3 Report
Dear Authors
I have read the resubmitted paper.
The manuscript still have some importan limitations
The topic is not novel and lacks of data about:
a) daily dose of antibiotics in the study period reletively to the infected patients;
b) classes of antibiotics used in infected patients and genus and species of the microrganisms responsible of infections.
Please provide to improve the manuscript according to my comments.
Author Response
Reply:
With regard to the objective of this study, it is important to stress the importance of the subject, since it is necessary to have a constant evaluation of surgical techniques, since these are modified over time and this can in turn influence the development of surgical site infections (SSI) and the costs derived from them. It should also be noted that a novel methodology has been used, since the Propensity Score has been used as a matching method, which, despite not being a new method, is innovative in its application in a study of economic costs.
a) Daily dose of antibiotics in the study period reletively to the infected patients;
Reply:
This question cannot be answered because the doses of antibiotics used in the treatment of surgical site infections (SSIs) were not one of the objectives of our work and therefore was not taken into account in the study design. During the study period, all patients who underwent cholecystectomy were followed for 30 days after surgery to verify whether or not they developed an SSI. In patients who developed an SSI, the SSI surveillance and prevention system also recorded the type of infection (superficial, deep or organ space), whether a surgical wound culture was requested, the type of microorganism isolated in the culture and the antibiotic used to treat the infection, but not the antibiotic dose. Not having the doses of antibiotics used is a limitation of our study since it contributes to underestimating the costs derived from SSIs.
b) Classes of antibiotics used in infected patients and genus and species of the microrganisms responsible of infections.
Please provide to improve the manuscript according to my comments.
Reply:
The most commonly used antibiotics in the treatment of SSIs after cholecystectomy were amoxicillin/clavulanic acid (34.55%; 38/110 patients), tazocel (28.18%; 31/110 patients) and meropenem (20.91%; 23/110 patients). In 14.55% (16/110) of patients with SSI, there were no data on the antibiotic treatment used.
As for the microorganisms responsible for SSIs, of the 110 patients who developed infection, 61.8% (68/110) had a surgical wound culture requested, which was positive in 88.2% (60/68). The pathogens most frequently identified in the microbiological cultures were Escherichia coli (35%; 21/60 cultures) and Enterococcus (30%; 18/60).

Round 2
Reviewer 3 Report
Dear Author
I read the revised version of the manuscript . It has been improved according to my suggestions.
I still have one request
Results line 196 Enterococcus Please specify the species ( faecium or faecalis or other) with the respective percentages
Author Response
Response:
Regarding the microorganisms responsible for SSIs, of the 110 patients who developed infection, 61.8% (68/110) had surgical wound cultures requested, being positive in 88.2% (60/68). The most frequently identified pathogens in the microbiological cultures were Escherichia coli (35%; 21/60 cultures) and Enterococcus (30%; 18/60). Regarding the genus Enterococcus, the species identified were E. faecalis (8/18) and E. faecium (5/18), in the remaining 5 cases the laboratory results did not indicate the species.

This manuscript is a resubmission of an earlier submission. The following is a list of the peer review reports and author responses from that submission.
Round 1
Reviewer 1 Report
This paper provides a clear and comprehensive overview of estimate the costs of SSIs in patients who underwent a cholecystectomy by a prospective observational cohort study. The study was well-designed and very meaningful. The revisions suggested below are mostly related to the results.
- The paper included 2,200 patients who had undergone a cholecystectomy, of which, 110
developed SSIs. But “99 cases were lost and so n = 2,101” in the table footnote. How many of those 99 cases developed SSIs?
- In table 3, this paper compared the hospital stay length between patients with developed SSIs and without developed SSIs. Were the sample of ‘Not exposed’those 110 patients after matching using the propensity score? In my opinion,the impact of SSIs on hospital stay length and cost was a little weak.
Reviewer 2 Report
This reviewer commends the authors for this interesting study.
Here are my comments
- This study has a major design flaw: From this study, it is not possible to make a generalization whether the cost and risk factors for post-cholecystectomy SSI’s and prolonged hospital stay are also applicable to other surgical procedures mentioned.
- Assuming that diverse types of major surgeries are undertaken in that health care facility, most major surgeries are more complex than a simple cholecystectomy and would be associated with a different set of risk factors and costs associated with a variably prolonged hospital stay.
- What exactly is the hypothesis/research question for this study? The authors state that they run a regression analysis on the variables. What is the variance explained by each independent factor toward the outcome variable of the cost associated with a prolonged hospital stay?
- If the authors really want to produce an evidence-based on factors associated with SSI infections to help their healthcare facility to design a strategic intervention plan to reduce SSI’s, cholecystectomy per se, a surgical procedure that takes about an hour or two to complete may not be the best choice and representative enough to give the big picture of SSI’s, preventable hospital stay, and associated cost savings in that health-care facility
- The most common surgical operations associated with SSI’s detected in 30-day surveillance include abdominal aortic aneurysm repair, limb amputations, appendix surgery, shunt for dialysis, biliary duct, liver or pancreatic surgery, carotid endarterectomy, gallbladder surgery, colon surgery, cesarean section, gastric surgery, heart transplant, abdominal hysterectomy, laminectomy, liver transplant, neck surgery, kidney surgery, ovarian surgery, prostate surgery, rectal surgery, small bowel surgery, spleen surgery, thoracic surgery, thyroid and/or parathyroid surgery, vaginal hysterectomy, and exploratory laparotomy.
- In conclusion, this study is analogous to “looking through a keyhole to try to find out what is going inside a large complex house.” This study does help us expand our understanding of SSIs and falls short of producing new evidence that will help the hospital/health care facility design an intervention strategy to reduce SSIs, prevent extended hospital stay, and/or associated costs.
Reference
National Healthcare Safety Network (2021). Surgical Site Infection Event (SSI). https://www.cdc.gov/nhsn/pdfs/pscmanual/9pscssicurrent.pdf
Reviewer 3 Report
This is a nice observational study on the economic costs (derived from enlenghtened hospital stay) of SSI after cholecystectomy
The study uses a rather sophisticated method of comparison but some basic assumptions are not clearly defined.
Mainly, authors should clarify:
- How the final model logistic model for the propensity scores (set of variables that best …) was fitted? Stepwise, manual,... Was the fitting of the model (Hosmer-Lemeshow test,…) for predicted versus observed cases tested?
- Which factors were actually included in the propensity score
- How was selected the matched cohort among the non-infected patients
- There is a common problem when comparing length of hospital stay due to the habitual non-normal distribution of these data which precludes the use of normal test such as the Student t test. When this assumption is not satisfied difference of means estimate could be biased. Were the data of postsurgical hospital stay normally distributed? Homogeneity of variances of the data was assessed? Were student t test appropriate for comparison of length of stay? Had not been preferable to compare medians instead of means? (median test)
Minor comments
Abstract
Statistical methods
“Chi-squared tests were used to determine the correlation of these factors with the development of an SSI” should say more precisely “Chi-squared tests were used to determine the association of these factors with the development of an SSI”
Results
It is noticeable that both groups statistically differed in the proportion of open surgery (Non laparoscopic) technique (47% of infected vs 14% of non-infected). This difference deserves discussion since open surgery is associated to a larger postsurgical stay comparing with laparoscopic surgery.
Discussion
This assert “Adopting the recommended strategies for HAI surveillance can reduce the incidence of SSIs to 204 50% [30]. Thus, their implementation and compliance with these measures should be prioritized in every healthcare system to reduce both morbidity and mortality and economic costs associated with SSIs.” is not derived from the study and should be removed.
Reviewer 4 Report
Dear Authors,
I read the submitted paper titled "Estimated costs associated with surgical site infections in patients undergoing cholecystectomy".
The topic is of interest, but material and methods and results must be improved .
I would like to underline some issues of concern which in my opinion need major revisions
Material and Methods
a) line86- 87 "We used the criteria established by the Centers for Disease Control and Prevention (CDC) to define and classify SSIs [17].
This reference is rather old. Please add a more recent reference
Results
b) line 142-143
"We included 2,200 patients who had undergone a cholecystectomy, of which, 110 (5.0%) developed SSIs."
Wich microrganisms were isolated in these infected patients? Please specifySpecify also the site of isolation ( Wound, drainage, blood) with their respective percentages c) Tab.1 and Tab. Antibiotic prophylaxis Please specify wich antibiotics have been used for prophylaxis